# Web Agents Are Still Greedy: Progress-Aware Action Generation and Selection via Meta-Plan

## Abstract

Despite recent advancements in the reasoning and planning capabilities of large language models that enable automated web agent tasks, state-of-the-art web agent frameworks still exhibit high task failure rates and lag behind human performance, hindering their deployment and generalization across diverse website environments. In this paper, we identify key limitations in these web agent frameworks, attributing failure to their greedy reasoning without an understanding of the current task progress regarding what key steps have been completed and what remains in the next steps. Due to this lack of progress awareness, existing web agent frameworks often fall into suboptimal behaviors such as skipping essential key steps and producing incoherent or oscillatory trajectories, which hinder task completion. To address these limitations, we propose MAPLE, a simple yet effective add-on method with **MetA-PL**an guided action generation and s**E**lection. Our proposed method equips existing web agent frameworks to self-reason with an explicit meta-plan that encompasses high-level sequential guidelines for solving the task, enabling them to keep track of current progress and consistently adhere to the given guidelines. Experiments across diverse website benchmarks demonstrate that MAPLE largely outperforms previous state-of-the-art web agent frameworks by addressing their common failures and suboptimal behaviors induced by the lack of progress awareness.

## 1 Introduction

Recent advancements in the reasoning capabilities of Large Language Models (LLMs) have shown versatility in a wide range of complex language reasoning tasks (Wei et al., 2022; Wang et al., 2022b; Liu et al., 2024), as well as agentic applications (Yao et al., 2023; Shinn et al., 2023) that require sequential interaction with environments. Among these, however, generalization on web agent tasks (Zhou et al., 2023; Koh et al., 2024a; Deng et al., 2023) still remains limited and lags behind human performance, due to the complexity of long-horizon planning and the dynamic nature of web environments which require interacting with a large number of graphical user interface (GUI) elements such as buttons and input fields. To better generalize on such complex website environments, several web agent frameworks (Koh et al., 2024b; Gu et al., 2024) have focused on adaptive control and planning strategies to cope with dynamically changing web observations in response to each action taken by the agent. Specifically, TreeSearch (Koh et al., 2024b) proposed a trial-and-error planning approach that allows agents to backtrack to previous states upon encountering failure, while WebDreamer (Gu et al., 2024) adopted simulation-based planning that imagines the expected outcomes of each plan and selects the one most aligned with the desired final goal.

However, these state-of-the-art (SOTA) frameworks still exhibit limited generalization on complex long-horizon tasks that require structured reasoning. In this paper, we identify the key limitations of these frameworks induced by their greedy reasoning (Yao et al., 2023) to predict the next action, without an understanding of the current progress regarding what key steps have been completed and what remains in the next steps. Due to this lack of progress awareness, they exhibit suboptimal behaviors such as skipping essential key steps and producing incoherent action trajectories. As illustrated in Figure 1, the agent greedily predicts next actions to achieve the goal without proper grounding on current progress, resulting in incoherent behavior such as scrolling down the page

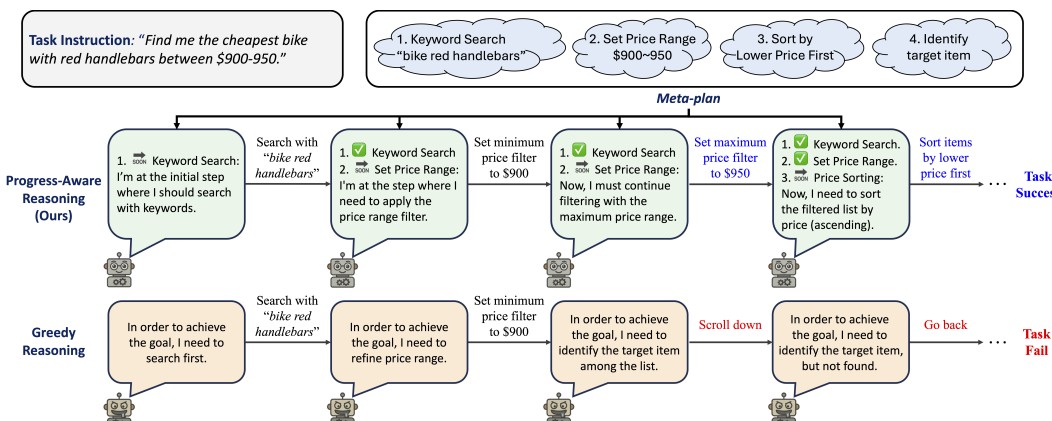

Figure 1: Agent trajectories under different reasoning approaches. Upper flow: progress-aware reasoning approach (ours). Lower flow: greedy reasoning (Gu et al., 2024; Koh et al., 2024b). Progress-aware reasoning via the meta-plan enables the agent to successfully complete the task by maintaining a coherent trajectory, without skipping essential steps.

where it is expected to coherently set the maximum price, and further skipping the essential sorting step required to identify the cheapest item.

To address these limitations, we propose MAPLE, which incorporates web agents with **M**et**A**-**PL**an guided action generation and s**E**lection to enable more faithful and coherent navigation. Specifically, we first establish a meta-plan where the given task instruction is decomposed into the high-level subtasks that are essential for achieving the goal. At each sequential step, the agent predicts the next action based on understanding of the current progress, specifically reasoning about which subtasks have been completed so far and which remain within the meta-plan. As shown in Figure 1 and Appendix E, this simple yet effective progress-aware reasoning addresses the common failures of SOTA web agents (Gu et al., 2024; Koh et al., 2024b), such as incoherent or oscillatory trajectories, premature termination, and invalid action repetition, by explicitly following the meta-plan that helps steer toward the given task goal.

Moreover, we note that such progress-aware reasoning can also address a key limitation of the action value function from SOTA web agents (Gu et al., 2024; Koh et al., 2024b), which quantifies the value of each action candidate to select the most promising one that can be taken on the current web observation. Specifically, conventional value functions tend to assign the same scores to all *on-track* action candidates that can plausibly serve as intermediate steps toward the goal, failing to identify a more goal-directed one from less effective yet valid on-track ones. For example, in tasks that involve finding the most recent item from the shopping listings, they may select inefficient, detouring on-track action (e.g., jumping to the oldest page and searching backwards) rather than the effective, straightforward action (e.g., searching from the latest page). This inefficiency hampers the agent's progress, causing unnecessary detours that impede convergence toward the objective. To address this, we devise a new action selection method that tracks current progress and explicitly votes for the most promising next action that aligns with the next remaining step within the meta-plan. Consequently, the agent is directed toward the objective by taking more effective on-track actions based on the guidance from the meta-plan.

Our comprehensive experiments on various website benchmarks from VisualWebArena dataset (Koh et al., 2024a) demonstrate that this straightforward meta-plan guided action generation and selection achieves superior task success rates compared to the previous SOTA web agent frameworks (Gu et al., 2024; Koh et al., 2024b). Also, integrating our methods on top of these frameworks further yields substantial performance gains, achieving +40% relative improvement for Webdreamer (Gu et al., 2024) in Classifieds benchmark. In summary, our contribution is threefold:

- We systematically reveal the key limitations of state-of-the-art web agent frameworks, where greedy reasoning without keeping track of the current progress status leads to sub-optimal trajectories.
- We propose meta-plan guided reasoning for action generation and selection that enables web agents to explicitly track their current progress and consistently take more goal-directed actions, promoting more faithful and coherent web navigation.

- Extensive experiments on various website benchmarks show that our method achieves superior performance compared to the previous state-of-the-art web agent frameworks, and even further enhances their performance when our methods are integrated into them.

## 2 RELATED WORK

### 2.1 WEB AGENTS

With the advancements in the reasoning capability of LLMs (Wei et al., 2022; Zhou et al., 2022), the seminal works (Gur et al., 2023; Yao et al., 2023; Deng et al., 2023) employed LLMs as a controller to automate web tasks by navigating within the web environments (Deng et al., 2023; Zhou et al., 2023; Koh et al., 2024a). Since processing an enormous amount of textual tokens from HTML sources is resource-inefficient with low information density, Cheng et al. (2024); Zheng et al. (2024); Lin et al. (2025) improved its efficiency by leveraging a smaller number of visual patch tokens from web screenshots based on multimodal LLMs (Achiam et al., 2023; Wang et al., 2024). Another line of work (Gu et al., 2024; Koh et al., 2024b) improved the planning ability of web agents to interact with a large number of GUI components from dynamic and complex web environments. However, these state-of-the-art web agent frameworks still exhibit limited generalization on long-horizon tasks that may require reasoning about the current progress, including what steps have been completed and what next steps remain. Due to the lack of such progress awareness, they often suffer from suboptimal behaviors such as skipping essential key steps for solving the long-horizon tasks. To address this, we address this limitation by guiding the web agent with a meta-plan to keep track of its current progress, promoting a more faithful and coherent web navigation.

### 2.2 AGENTS GUIDED BY META-PLAN

To effectively solve the complex and long-horizon tasks, recent studies have focused on a plan-and-solve strategy via guidance from a meta-plan that contains high-level sequential concepts to solve the given task. Specifically, Khot et al. (2022); Zheng et al. (2023); Wang et al. (2023) simplified complex language reasoning tasks with the meta-plan by decomposing the given task into simpler language subtasks. Also, agentic frameworks such as TPTU (Ruan et al., 2023), HuggingGPT (Shen et al., 2023), and LLMCompiler (Kim et al., 2024) leveraged meta-plan to effectively coordinate diverse agentic tools, such as SQL database querying. Another line of work (Erdogan et al., 2025; Xiong et al., 2025) explicitly integrated the meta-plan into web agent frameworks where the agent's action policy is explicitly governed by the high-level guidelines from the meta-plan. Plan-and-Act (Erdogan et al., 2025) developed a more sophisticated meta-plan by training the meta-planner on extra synthetic data and enabling it to adapt to unforeseen variations in dynamic web environments. Similarly, MPO (Xiong et al., 2025) also optimized the meta-planner via feedback from the agent during web navigation. Despite the advances, these frameworks may still suffer from potential plan-to-action grounding errors that may arise when the high-level meta-plan is incorrectly decoded into a single low-level action, as abstract concepts within the meta-plan are often misinterpreted by the policy model. Consequently, these grounding errors can lead to suboptimal trajectories. In this paper, we mitigate such errors by generating multiple candidate actions from the policy model and selecting the most promising one among the candidates via our proposed action evaluator, based on its alignment with the next intended step in the meta-plan.

## 3 METHOD

In this section, we first describe the problem formulation of existing web agents (Sec 3.1) and then introduce MAPLE, a general add-on method designed to enhance the existing web agents via progress-aware action generation and selection. Specifically, given a web-based task, we propose two key components for improving agent reliability and coherence: (i) Action Candidate Generation with Meta-Plan (Sec 3.2), where the task is decomposed into several high-level guidelines that enable the agent to keep track of its current progress when predicting the next action candidates, and (ii) Action Selection with Meta-Plan (Sec 3.3), which opts for the most goal-aligned action among the candidates based on its relevance to the next intended step within the meta-plan. Through these two components, MAPLE enables web agents to follow a more structured and coherent execution path, mitigating common failures such as skipping essential steps and trajectory incoherence.

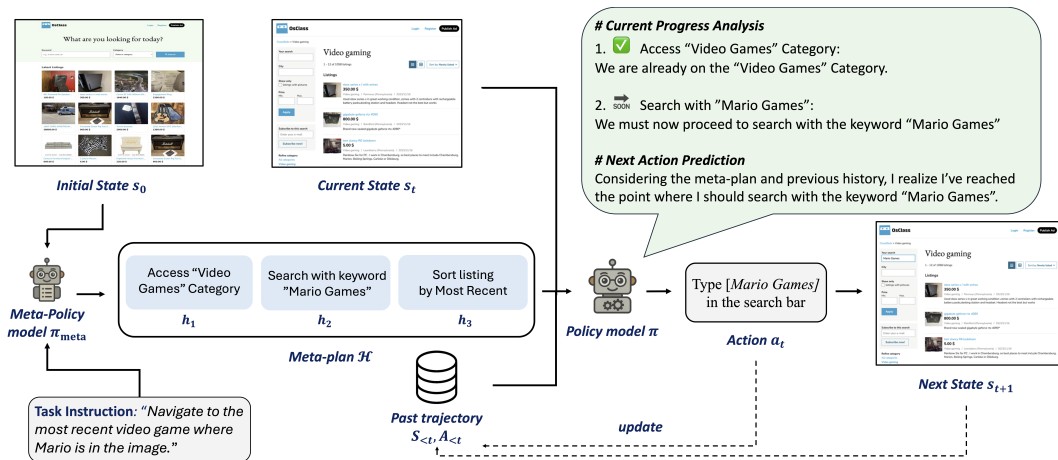

Figure 2: Overview of action generation governed by meta-plan. Based on the progress-aware reasoning, policy model generates coherent action aligned with upcoming steps in the meta-plan.

## 3.1 PRELIMINARY

In the standard web agent framework, an agent aims to complete a given instruction $I$ (e.g., "*Find me the cheapest bike with red handlebars between $900-950*") through sequential interactions with a dynamic web environment. At each timestep $t$, given the current observation $s_t$ including an HTML document and the corresponding screenshot image, the agent predicts the next action $a_t$ (e.g., "*type 'bike' in the search bar*") via a policy model $\pi$ powered by multi-modal LLM:

$$a_t = \pi(s_t, I). \tag{1}$$

Under the Markov Decision Process (MDP), the policy $\pi$ aims to generate an *on-track* action that transitions to the desired next state $s_{t+1}$, which possibly steers the agent toward the final state $s_{\hat{t}}$ that satisfies the objective $I$:

$$s_{t+1} = \mathcal{S}(s_t, a_t), \quad \text{where } \exists \ \hat{t} \geq t+1 \ \text{ s.t. } R(s_{\hat{t}} \mid I) = 1. \tag{2}$$

Here, $\mathcal{S}$ denotes the website transition function and $R(\cdot)$ is a binary indicator of whether the final state $s_{\hat{t}}$ fulfills the given objective $I$ or not. However, existing frameworks (Gu et al., 2024; Koh et al., 2024b) often fall into a suboptimal state $s_{t+1}^*$ due to its greedy reasoning process of $\pi$. Specifically, for each time step $t$, the policy model $\pi$ only reasons about the immediate next plan for achieving the goal under the current state $s_t$ without explicit consideration of the past trajectory $\boldsymbol{S}_{<t} = (s_0, \ldots, s_{t-1})$ and $\boldsymbol{A}_{<t} = (a_0, \ldots, a_{t-1})$, predicting actions without adequately reflecting on what has been achieved so far and what remains in the task (Figure 1). This lack of progress-aware reasoning leads to three failure modes: 1) a suboptimal trajectory that omits essential key steps (e.g., sorting items by price for the task finding the cheapest item), 2) an incoherent trajectory that discontinues the progress achieved by prior on-track actions (e.g., scroll down action after setting minimum price, instead of coherently setting maximum price), and 3) an invalid repetitive behavior without completing the objective (e.g., repeating the scroll down action forever).

## 3.2 ACTION CANDIDATE GENERATION WITH META-PLAN

To promote more coherent and goal-aligned behavior, we explicitly ground the policy model $\pi$ on a meta-plan $\mathcal{H} = (h_1, h_2, \ldots, h_K)$, where each $h_k$ denotes a high-level subtask that must be completed in sequence to fulfill the objective $I$, as illustrated in Figure 2:

$$\mathcal{H} = \pi_{\text{meta}}(I, s_0). \tag{3}$$

Here, $\pi_{\text{meta}}$ denotes a meta-policy model which decomposes the objective $I$ into a sequence of high-level subtask plans $h_k$ starting from the initial state $s_0$. In practice, we employ the off-the-shelf multi-modal LLM (Hurst et al., 2024) as the meta-policy model to generate up to $K = 5$ subtasks (See Appendix A for the full prompt and example outputs). This meta-plan serves as a guide for the agent to keep track of its own progress and instills a structured vision of remaining subgoals. To ground the agent's next action based on current progress within the meta-plan, we revise Eq. 1

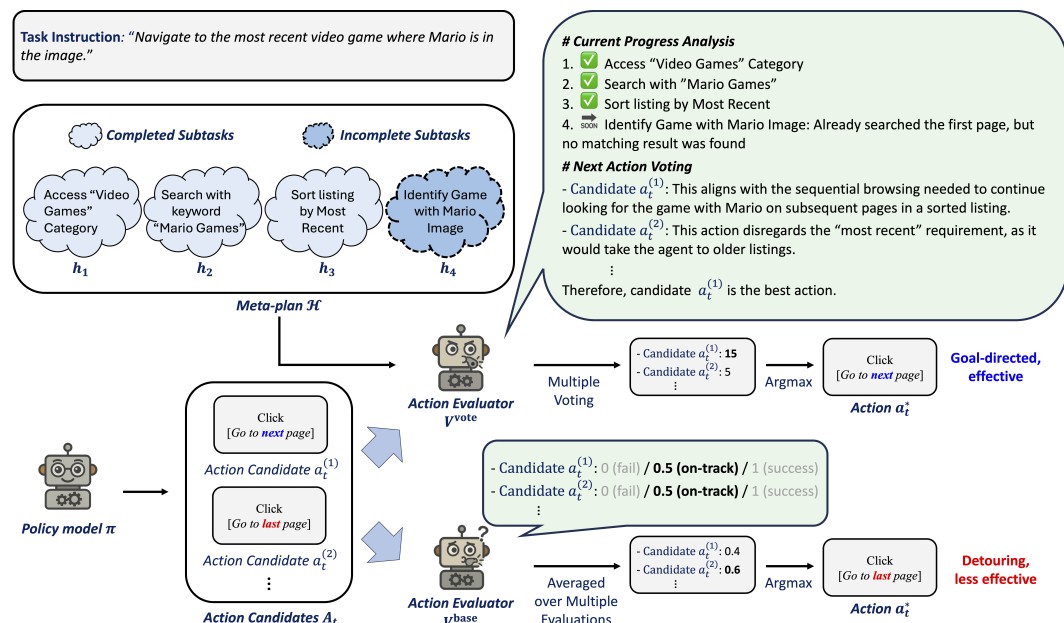

Figure 3: Overview of action selection governed by the meta-plan. Without guidance from the meta-plan (bottom), the conventional evaluator may indiscriminately select on-track yet detouring action (e.g., backward search despite the listings already being sorted by most recent first). In contrast, our evaluator (top), guided by progress-aware reasoning, explicitly votes for the effective on-track action (e.g., forward search) aligned with the next remaining subgoal in the meta-plan, thereby steering the agent onto a goal-directed trajectory.

for the policy model $\pi$ to be conditioned additionally on the past trajectory $(\boldsymbol{S}_{<t}, \boldsymbol{A}_{<t})$, and the meta-plan $\mathcal{H}$:

$$a_t = \pi(s_t, I \mid \boldsymbol{S}_{<t}, \boldsymbol{A}_{<t}, \mathcal{H}). \tag{4}$$

Under this reformulation, the policy model $\pi$ first analyzes current progress within the meta-plan by inspecting the past trajectory, and then predicts the next corresponding action candidate to achieve the remaining incomplete subtasks within the meta-plan. Specifically, for the policy model to self-reason on the current progress at each time step $t$, we define a binary completion variable $c_k \in \{0, 1\}$ for each subtask $h_k$, resulting in a progress vector $\boldsymbol{c}_t = [c_1, c_2, \ldots, c_K]$ where each $c_k = 1$ indicates that $h_k$ is considered completed:

$$\boldsymbol{c}_t = [\, \mathcal{C}(h_k \mid \boldsymbol{S}_{\leq t}, \boldsymbol{A}_{<t}) \,]_{k=1}^K \tag{5}$$

where $\mathcal{C}(\cdot)$ denotes judgment from the policy model whether each subtask has been completed by inspecting the action history $\boldsymbol{A}_{<t}$ and the corresponding observation history $\boldsymbol{S}_{\leq t} = (s_0, \ldots, s_t)$. To operationalize $\mathcal{C}(\cdot)$, we provide the policy model $\pi$ with few-shot reasoning examples that demonstrate how to infer completion status $c_k$ for each subtask $h_k$ (See Appendix B.2 for the full reasoning example). Using $\boldsymbol{c}_t$, the policy model then identifies the next upcoming incomplete subtask $h_{k^\star}$ where $c_1 = 1, \ldots, c_{k^\star-1} = 1$, and $c_{k^\star} = 0$. Finally, the policy model generates the next corresponding action candidate that can possibly complete $h_{k^\star}$. By conditioning the action candidate generation process on this progress-aware context, the agent avoids falling into the suboptimal, redundant, or oscillatory trajectories and maintains a coherent trajectory to achieve the given objective (Figure 1 and Appendix E).

### 3.3 Action Selection with Meta-Plan

For more reliable navigation of web agent, the policy model generates multiple action candidates available at each time step, and then the best candidate is selected via action value function $V^{\text{base}}(\cdot)$:

$$\mathcal{A}_t = \{a_t^{(1)}, \ldots, a_t^{(N)}\} \sim \pi(s_t, I \mid \boldsymbol{S}_{<t}, \boldsymbol{A}_{<t}, \mathcal{H}; \tau_g), \tag{6}$$

$$a_t^* = \arg\max_{a \in \mathcal{A}_t} \frac{1}{M} \sum_{m=1}^M V^{\text{base}}(s_t, a; \tau_e) \tag{7}$$

where $N$ and $M$ denote the number of generated action candidates and action evaluation rounds, respectively, while $\tau_g$ and $\tau_e$ are temperature parameters for $\pi$ and $V^{\text{base}}$. Although the generated candidates are considered *on-track* as they generally follow the meta-plan $\mathcal{H}$ and thus maintain progress toward the objective, not all on-track candidates are equally effective. Some actions may advance the agent more directly toward next subgoal in the meta-plan, while others may take detours and result in slower progress. However, conventional action value function $V^{\text{base}}$ (Gu et al., 2024; Koh et al., 2024b) evaluates each action candidate using a coarse rubric, assigning a score of 1.0 to actions that may finally complete the task and 0 to those likely to fail, while treating all the candidates deemed on-track equally with a score of 0.5 regardless of how effectively each action candidate advances the agent toward the goal. Although scores are averaged across multiple evaluations, such value functions often fail to identify the most effective on-track action, resulting in suboptimal decisions and ineffective trajectories (Figure 3). One might naively attempt to mitigate this by increasing granularity of scoring scheme to distinguish among on-track actions by effectiveness, but defining such a rubric for the action value function without a concrete criterion (e.g., "*give higher value to more effective on-track actions while lower values to less effective ones*") can be subjective and may lead to unreliable and inconsistent evaluations. To address this, instead of relying on the quantification of each action candidate without a certain criterion, we explicitly select the most effective on-track action by majority voting (Wang et al., 2022a) for the one most aligned with the next remaining subgoal in the meta-plan, which helps progress towards achieving the final objective:

$$a_t^* = \arg\max_{a \in \mathcal{A}_t} \sum_{m=1}^{M} \mathbb{I}\left(a = V_m^{\text{vote}}(s_t, \mathcal{A}_t \mid \boldsymbol{S}_{<t}, \boldsymbol{A}_{<t}, \mathcal{H}; \tau_e)\right) \tag{8}$$

where $\mathbb{I}(\cdot)$ denotes a binary indicator of whether the candidate $a$ was selected by the voting model $V_m^{\text{vote}}(\cdot)$ in the $m$-th voting round. For the voting model $V_m^{\text{vote}}$, we employ the off-the-shelf multi-modal LLM (Hurst et al., 2024) by default. $M$ and $\tau_e$ denote the number of voting rounds and the temperature for voting, respectively, with $\tau_e$ set to 1 by default to encourage diverse reasoning across voting rounds. In each round, $V_m^{\text{vote}}(\cdot)$ first identifies the next upcoming incomplete subgoal $h_{k^\star}$ where $c_1 = 1, \ldots, c_{k^\star-1} = 1$, and $c_{k^\star} = 0$, as in Sec. 3.2. Subsequently, it analyzes all the action candidates in terms of how effectively each contributes to achieving $h_{k^\star}$, and then votes for the most suitable one (See Appendix C for the full prompt and few-shot reasoning example). This progress-aware action selection guided by the meta-plan empowers the agent to identify a more goal-directed and effective trajectory toward completing the task. As shown in Figure 7 and Figure 8, our proposed action selection is also effective for addressing other common failure modes of the conventional value function, such as premature termination and invalid action repetition.

## 4 EXPERIMENTS

**Dataset.** We evaluate MAPLE on the widely used web agent benchmark, VisualWebArena (Koh et al., 2024a) (VWA) dataset to compare the performance against SOTA web agent frameworks. VWA consists of 910 tasks across three website environment subsets: Classifieds, Reddit, and Shopping. All the tasks are designed for evaluating the multi-modal agents, which require visual understanding of the webpage contents. Following Zheng et al. (2024); Gu et al. (2024); Koh et al. (2024b), we augmented the webpage screenshot with Set-of-Mark (Yang et al., 2023) prompting, which allows each HTML element to be grounded by its id when generating action candidates. For the evaluation metric, the success rate is calculated as the proportion of successfully completed tasks (e.g., finally navigated url matches the desired web url) to the entire number of tasks.

**Implementation details.** When generating action candidates for each time step, the policy $\pi$ is provided with in-context examples that demonstrate how to capture current progress within the meta-plan and predict the corresponding next action (See Appendix B.2). We set the maximum number of generated action candidates $N = 20$ and use nucleus sampling (Holtzman et al., 2019) with top-$p$ of 0.95 and temperature $\tau_g = 1.0$ in Eq. 6, unless specified. Among the generated action candidates, we feed top-5 frequently generated actions to our action selection (Eq. 8) to vote for the best action $a_t^*$. For the voting model $V^{\text{vote}}$, we provide in-context examples that demonstrate how to select the most promising action based on current progress (See Appendix C.2). Also, we set the temperature $\tau_e = 1.0$ and the number of voting rounds $M = 20$, unless specified. The agent is forced to stop execution after a maximum of 30 time steps, while treating as a failure if the same actions are

Table 1: Performance comparison with state-of-the-art web agent frameworks on VWA (Koh et al., 2024a). Values in the parentheses (△) denote relative improvement compared to the vanilla method, ReAct (Yao et al., 2023). For fair comparisons, we used GPT-4o for all the methods.

| Method | Action Gen. w/ Meta-Plan? | Action Select w/ Meta-Plan? | Success Rate (SR) ↑ | | | Total SR (△) ↑ |
|---|---|---|---|---|---|---|
| | | | Classifieds | Reddit | Shopping | |
| ReAct (Yao et al., 2023) | ✗ | ✗ | 17.9 | 14.3 | 19.3 | 17.6 (+0.0%) |
| ICAL (Sarch et al., 2024) | ✗ | ✗ | - | - | - | 23.4 (+33.0%) |
| AdaptAgent (Verma et al., 2024) | ✗ | ✗ | - | - | - | 23.9 (+35.8%) |
| GenericAgent (Chezelles et al., 2024) | ✗ | ✗ | - | - | - | 26.7 (+51.7%) |
| WebDreamer (Gu et al., 2024) | ✗ | ✗ | 23.2 | 17.5 | 26.3 | 23.2 (+31.8%) |
| TreeSearch (Koh et al., 2024b) | ✗ | ✗ | 26.8 | 20.6 | 28.9 | 26.2 (+48.9%) |
| MAPLE (Eq 6, 7) | ✓ | ✗ | 29.9 | 24.8 | 33.9 | 30.8 (+75.0%) |
| **MAPLE (Eq 6, 8)** | ✓ | ✓ | **32.1** | **26.7** | **36.9** | **33.3 (+89.2%)** |

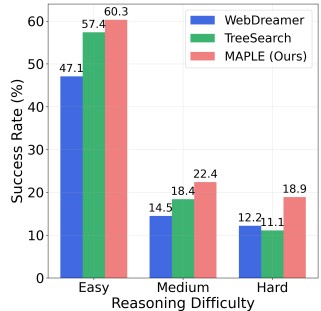

Figure 4: Comparison of task success rates depending on reasoning difficulty level in Classifieds subset of VWA.

Table 2: Add-on effect of the proposed methods on SOTA frameworks, WebDreamer and TreeSearch. Values in the parentheses (△) denote the relative improvement compared to each baseline framework.

| Method | Action Gen. w/ Meta-Plan? | Action Select w/ Meta-Plan? | Classifieds SR (△) ↑ |
|---|---|---|---|
| WebDreamer (Gu et al., 2024) | ✗ | ✗ | 23.2 (+0.0%) |
| + MAPLE (Eq 6, 7) | ✓ | ✗ | 29.9 (+28.9%) |
| **+ MAPLE (Eq 6, 8)** | ✓ | ✓ | **32.5 (+40.1%)** |
| TreeSearch (Koh et al., 2024b) | ✗ | ✗ | 26.8 (+0.0%) |
| + MAPLE (Eq 6, 7) | ✓ | ✗ | 28.6 (+6.7%) |
| **+ MAPLE (Eq 6, 8)** | ✓ | ✓ | **29.9 (+11.6%)** |

repeated for 5 consecutive steps (Gu et al., 2024). For all experiments, we employed GPT-4o (Hurst et al., 2024) for the policy model $\pi$, the meta-policy model $\pi_{\text{meta}}$, and the action evaluator $V^{\text{vote}}$.

## 4.1 MAIN RESULTS

**Comparison with SOTA frameworks.** In Table 1, we compare MAPLE with other state-of-the-art web agent frameworks on VWA benchmark. The vanilla method, ReAct, shows a low success rate across all the subsets due to its greedy reasoning without tracking current progress during the action generation process, while recent baselines enhanced the performance via human feedback (ICAL and AdaptAgent) and sophisticated planning strategies with simulation- or tree-based exploration (WebDreamer and TreeSearch). However, MAPLE consistently exhibits the best success rates on all the subsets, largely outperforming all the baselines. Specifically, when the action generation process is guided by the meta-plan as in Eq. 6, it already outperforms the previous SOTA GenericAgent by a large margin, achieving +4.1% absolute points in total. Also, applying the action selection via meta-plan (Eq 8) brings further improvement of +6.6% absolute points over GenericAgent, while achieving +89% relative improvement compared to the vanilla method.

In Figure 4, we further analyze the performance across different levels of reasoning difficulty. Notably, the baselines such as WebDreamer and TreeSearch substantially degrade their performance as the reasoning difficulty increases. Due to the greedy reasoning that lacks awareness of current progress, they exhibit suboptimal behavior such as skipping essential key steps in the hard tasks that require long-horizon planning (Figure 1). However, MAPLE shows significantly improved performance on hard tasks, achieving +70% relative improvement compared to TreeSearch. These results demonstrate the effectiveness of our progress-aware reasoning for action generation and selection via meta-plan, by steering away from falling into the suboptimal trajectories and instead pursuing a more reliable and coherent trajectory for completing the long-horizon tasks.

Table 3: Ablation study of progress-aware reasoning (PAR) and meta-plan for action generation, evaluated on Classifieds depending on the level of reasoning difficulty. Values in the parentheses ($\triangle$) denote the relative overall improvement compared to the baseline method (Gu et al., 2024) without progress-aware reasoning.

| PAR? | PAR w/ Meta-plan? | Reasoning difficulty (SR) ↑ | | | Classifieds SR ($\triangle$) ↑ |
|---|---|---|---|---|---|
| | | Easy | Medium | Hard | |
| ✗ | ✗ | 47.1 | 14.5 | 12.2 | 23.2 (+0.0%) |
| ✓ | ✗ | 51.5 | 18.4 | 15.6 | 26.9 (+15.9%) |
| ✓ | ✓ | **55.9** | **22.4** | **16.7** | **29.9 (+28.9%)** |

Table 4: Effect of dynamic replanning of meta-plan (Erdogan et al., 2025) on Classifieds, depending on the level of reasoning difficulty. Values in the parentheses ($\triangle$) denote the relative overall improvement compared to the baseline method (Gu et al., 2024) without the meta-plan guided action generation and dynamic replanning.

| Action Gen. w/ Meta-Plan? | Dynamic Replanning? | Reasoning difficulty (SR) ↑ | | | Classifieds SR ($\triangle$) ↑ |
|---|---|---|---|---|---|
| | | Easy | Medium | Hard | |
| ✗ | ✗ | 47.1 | 14.5 | 12.2 | 23.2 (+0.0%) |
| ✓ | ✗ | **55.9** | **22.4** | **16.7** | **29.9 (+28.9%)** |
| ✓ | ✓ | 55.9 | 21.1 | 14.4 | 28.6 (+23.3%) |

(a) $N$ candidates ($\tau_g = 1.0$)  (b) Temperature $\tau_g$ ($N = 20$)  (c) $M$ voting ($\tau_e = 1.0$)  (d) Temperature $\tau_e$ ($M = 20$)

Figure 5: Ablation study on hyperparameters in meta-plan guided action generation and selection. (a), (b): Effects of varying the number of action candidates $N$ and temperature $\tau_g$ in action generation, respectively. (c), (d): Effects of varying the number of voting rounds $M$ and temperature $\tau_e$ in action selection, respectively. Red dotted lines indicate the performance of the baseline method (Gu et al., 2024) without meta-plan guidance.

**Plug-and-play with SOTA frameworks.** Since our proposed methods can be easily integrated into any prompt-based web agent frameworks via few-shot reasoning examples, we further investigate the add-on effect of MAPLE into the state-of-the-art frameworks, WebDreamer and TreeSearch. In Table 2, both action generation and selection via meta-plan consistently improve the success rates of all the baseline frameworks, yielding cumulative performance gains. Specifically, our action generation brings a remarkable +29% relative improvement to WebDreamer, while our action selection further contributes an additional +40% relative improvement. These results further corroborate that MAPLE is not only effective as a standalone framework but also serves as a generalizable plug-and-play module that resolves common failure modes (Appendix E) in conventional frameworks, thereby yielding substantial performance gains.

## 4.2 ABLATION STUDIES AND ANALYSIS

We conduct ablation studies and detailed analysis to assess the effectiveness of each component and the sensitivity of the hyperparameters comprising MAPLE. Unless specified, we applied both meta-plan guided action generation (Eq. 6) and selection (Eq. 8). Also, we report task success rate (SR) on Classifieds subset in VWA benchmark (Koh et al., 2024a).

**Effect of progress-aware reasoning.** In Table 3, we analyze the effect of progress-aware reasoning (Eq. 5) when generating next action candidates. Interestingly, equipping the policy model $\pi$ with the ability to naively reason about the agent's current progress by only analyzing history of actions $A_{<t}$ and observations $S_{\leq t}$, even without the meta-plan $\mathcal{H}$, substantially improves the performance over the vanilla greedy action generation that lacks such progress-aware reasoning. Moreover, when the policy model $\pi$ is further conditioned on the meta-plan to reason about the current progress and to generate next actions coherent with the remaining steps of the meta-plan (Sec 3.2), the success rate is additionally enhanced, up to 29% of relative improvement from the baseline. We also observe that our progress-aware reasoning is effective in resolving challenging failure cases that involve falling into incoherent or oscillatory trajectories, as shown in Figure 1 (See Appendix E for more qualitative examples). These results highlight that the progress awareness via meta-plan is essential for promoting web agents to navigate under a coherent and goal-directed trajectory.

**Effect of $N$ and $\tau_g$ for action generation.** For generating action candidates that adhere to the meta-plan as in Eq. 6, we investigate the effect of the number of generated action candidates $N$ and the temperature $\tau_g$ when decoding each candidate. In Figure 5a, for all the choices of $N$, our proposed action generation consistently brings improved performance compared to the baseline case without meta-plan guidance. We observe that the success rate increases monotonically as $N$ grows up to 20, while further increasing $N$ up to 40 only entails $2\times$ increased computation of output tokens without commensurate performance gains. Therefore, we fix $N = 20$ as a default to ensure a solid success rate while maintaining computational efficiency. We also note that leveraging only a single action candidate (i.e., $N = 1$) as in Erdogan et al. (2025), which obviates the need for the action selection process (Sec. 3.3), leaves substantial room for improvement compared to $N > 1$ cases where the proposed action selection is applied. This result possibly suggests that relying on a single decoded action may not be robust to potential errors in plan-to-act grounding, as the abstract concepts in the meta-plan are often misinterpreted by the action generator. We mitigate such errors by generating multiple action candidates and introducing another verifier (Eq. 8) that selects the most promising action based on the alignment with the meta-plan. In Figure 5b, we observe a similar trend where all the choices of the temperature $\tau_g$ consistently exhibit improved performance compared to the baseline. Notably, the success rate gradually increases as $\tau_g$ grows up to 1, highlighting the importance of promoting reasoning diversity for broadening the set of promising action candidates.

**Effect of $M$ and $\tau_e$ for action selection.** To identify the most promising action candidate that adheres to the meta-plan as in Eq. 8, we investigate the effect of the number of voting rounds $M$ and the temperature $\tau_e$ in Figure 5c and 5d, respectively. Notably, our proposed action selection consistently delivers performance improvement to the baseline case without meta-plan guidance during action selection, regardless of the choices of both $M$ and $\tau_e$. Similar to the trend shown in the parameter study of action generation (Figure 5a, 5b), we observe that the performance is saturated at $M = 20$ to achieve the best performance. Moreover, increasing the temperature $\tau_e$ monotonically improves the success rate. This result further corroborates the importance of incorporating diverse perspectives in the voting process to identify the most promising action aligned with the meta-plan.

**Dynamic replanning of meta-plan.** To cope with unexpected variations during web navigation, Erdogan et al. (2025) proposed dynamically updating the meta-plan rather than relying on a static one established at the initial state. Following Erdogan et al. (2025), we further investigated the effect of dynamic replanning by updating the meta-plan after every action execution and generating subsequent actions based on the revised meta-plan. For updating the meta-plan, we prompted the meta-policy model to revise or remove parts of the remaining incomplete plan that are no longer executable for completing the task (See Appendix D for the full prompt). The results in Table 4 show that action generation guided by meta-plan consistently exhibits improved performance compared to the baseline, while further applying dynamic replanning leads to a slight performance degradation on medium- and hard-level reasoning tasks. This result possibly indicates that frequent updates of the meta-plan can rather lead to the suboptimal trajectories in the long-horizon tasks, such as generating incoherent actions that disrupt the continuity of the progress achieved by the previous meta-plan, or skipping essential steps due to premature removal of necessary plans (Figure 1).

## 5 CONCLUSION

In this paper, we identify the key limitations of existing state-of-the-art web agent frameworks where the greedy reasoning without progress awareness often leads to suboptimal behaviors, such as skipping essential procedures and generating incoherent trajectories, hampering the successful completion of the web tasks. To address this, we propose the progress-aware reasoning approach to align the next action with a meta-plan that encompasses a sequence of high-level guidelines, allowing the agent to monitor its current progress and consistently adhere to the given guidelines. Also, we further revisit the fundamental limitations of existing action evaluation mechanisms where suboptimal actions are often assigned higher values and hence selected. To remedy this, we introduce a new action evaluation method by majority voting based on the meta-plan, enabling the agent to consistently select more effective actions that direct toward the goal without falling into detouring trajectories. Through extensive experiments across diverse web benchmarks and rigorous ablation studies, we demonstrate that our progress-aware action generation and selection methods can resolve the common failure cases frequently encountered by the conventional web agent frameworks, achieving the state-of-the-art performance while serving as an effective plug-in module.

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

APPENDIX

## A  META-PLAN GENERATION

In this section, we provide the full prompt for generating the meta-plan in Sec 3.2 as below:

---

**System Prompt**

You are an expert web navigation planner. Given a user's task intent, your job is to break down the task into a sequence of essential stages that must be completed to achieve the task. These stages will be used to evaluate an agent's progress.

For each stage, provide:
1. A short but descriptive name for the stage
2. A detailed description of what should be accomplished in this stage

The stages should form a logical progression from start to completion of the task.
The number of stages should be adjusted according to the complexity of the task, typically ranging from 3 to 5 stages.

Format your response as a JSON array of stage objects without additional commentary.
Example:

```
[
    {
        "stage_name": "Search",
        "description": "Search for the target product
        using appropriate keywords"
    },
    {
        "stage_name": "Price Comparison",
        "description": "Compare prices, potentially
        using sorting or filtering functionality"
    },
    ...
]
```

---

**Current Inputs**

Current webscreen page screenshot: [INITIAL WEBPAGE SCREENSHOT IMAGE]
Define the essential sequence of stages for this web task: [TASK INSTRUCTION]

---

The example generated meta-plans are given as follows. We reformatted JSON array for visibility:

---

**TASK: Find me the cheapest bike with red handlebars between $900-950**

1. `Select Category` — Click on the 'Bikes' category from the category section.

2. `Keyword Search` — Enter 'red handlebars' into the 'Keyword' search box.

3. `Set Price Range` — Set the price range to $900-$950 using any available price filter options.

4. `Search for Bikes` — Click on the 'Search' button to apply the filters and search for bikes with red handlebars within the specified price range.

---

5. `Sort by Price` — Sort the search results by price in ascending order to find the cheapest bike.

---

**TASK: Find the most recently listed coffee maker with a touch screen. Add a 5 star rating with title "Great item" and text "Would recommend!"**

1. `Search for Coffee Maker` — Enter 'coffee maker' into the keyword search bar and select the 'Appliances' category from the dropdown menu. Then click the 'Search' button.

2. `Sort by Latest Listings` — Once the search results are displayed, sort them by the most recently listed items to find the newest coffee maker.

3. `Select Coffee Maker with Touch Screen` — Browse the sorted listings to find a coffee maker that specifically mentions having a touch screen in its description, title, or image, and then click on it to view the details.

4. `Add Rating and Review` — On the coffee maker's detail page, locate the section for adding a review. Provide a 5-star rating, and enter the title 'Great item' and the text 'Would recommend!' for your review.

5. `Submit Review` — Submit your review to finalize the rating and feedback for the coffee maker.

## B  ACTION CANDIDATE GENERATION

In this section, we provide the full prompt with few-shot reasoning example for our proposed action candidate generation in Sec 3.2.

### B.1  SYSTEM PROMPT

---

**System Prompt**

You are an autonomous intelligent agent tasked with navigating a web browser. You will be given web-based tasks. These tasks will be accomplished through the use of specific actions you can issue.
Here's the information you'll have:

- **User's objective:** This is the task you're trying to complete.

- **Current URL:** This is the page you're currently navigating.

- **Current screenshot:** A screenshot of the current webpage, with each interactable element assigned a unique numerical id.

- **Current Observation:** Lists the IDs of all interactable elements in the format `[id] [tagType] [text content]`. `tagType` is the type of the element, such as button, link, or textbox.

- **Open tabs:** The tabs you have opened.

- **Previous actions:** History of actions that you have performed. It may be helpful to track your progress.

- **Previous screenshots:** History of previous screenshots leading up to the current state by previous actions history. It may be helpful to track your progress.

- **Meta-plan:** A structured high-level guideline outlining the logical sequence of steps you should consistently follow to effectively solve any given task.

**Actions you can perform:**

---

- *Page Operations:* `click [id]`, `type [id] [content]`, `hover [id]`, `scroll [down]`, etc.
- *Tab Management:* `new_tab`, `tab_focus [index]`, `close_tab`
- *Navigation:* `goto [url]`, `go_back`, `go_forward`
- *Completion:* `stop [answer]`

**Rules to Follow:**

1. Only issue valid action from the current observation.

2. Issue one action at a time.

3. Follow the examples to reason step by step and then issue the next action.

4. Based on the meta-plan, explicitly identify your current stage by analyzing previous history and the current observation. Then, reason clearly about the immediate next step required to complete the next unfinished stage in the meta-plan, as part of a coherent plan to ultimately achieve the objective.

5. Output action in the form: `In summary, the next action I will perform is 'click [1234]'`.

6. Use `stop [answer]` when you think the task is completed.

## B.2   FEW-SHOT REASONING EXAMPLES

**Few-shot Reasoning Examples**

**## Inputs**

**Objective:** Help me find the cheapest dark colored guitar.
**Current URL:** `http://_CLASSIFIEDS_DOMAIN_`
**Current screenshot:**

**Current Observation:**

```
[...]
[5] [INPUT] []
[6] [SELECT] [Select a category]
[7] [BUTTON] [Search]
[9] [A] [JBL Powered PA Speaker w/ Stand]
[11] [A] [xbox series x / with extras]
[...]
```

**Open tabs:** [Classifieds]

**Previous actions:** None
**Previous screenshots:** None
**Meta-plan:**

- 1. Navigate to classifieds website: Access the classifieds website where musical instruments are sold.

- 2. Search for guitar listings: Use the search feature by entering 'guitar' to display relevant listings.

- 3. Sort by price: Sort the guitar listings in ascending order of price.

- 4. Filter by Color and Price: Review the sorted guitar listings to identify guitars that are dark-colored and note their prices.

- 5. Select Cheapest Guitar: Compare identified dark-colored guitars and choose the listing with the lowest price.

---

## Output

**Reasoning:**

- *Step 1 (Task Analysis):* The objective clearly asks for the cheapest dark-colored guitar available on this site.

- *Step 2 (Current Progress Analysis):*

    ✓ 1. Navigate to classifieds website — The user is already on the homepage of the website.

    ✗ 2. Search for guitar listings — This is the next required step.

    Given that I have successfully completed the first stage of the meta-plan (i.e., accessing the classifieds website), the next action should be to initiate the keyword search, as outlined in the second stage of the meta-plan.

- *Step 3 (Current Observation Analysis):* I observe a search box (id [5]) which allows searching for specific items.

- *Step 4 (Predict next action):* We should type "guitar" into the search box.

**Action:** `In summary, the next action I will perform is 'type [5] guitar\n'`

## B.3 CURRENT INPUTS

**Current Inputs**

**Objective:** [TASK INSTRUCTION]
**Current URL:** [CURRENT URL]
**Current screenshot:** [CURRENT WEBPAGE SCREENSHOT IMAGE]
**Current Observation:** [CURRENT OBSERVATION]
**Open tabs:** [OPEN TABS]
**Previous actions:** [PREVIOUS ACTIONS]
**Previous screenshots:** [PREVIOUS SCREENSHOTS]
**Meta-plan:** [META-PLAN]

## C  ACTION SELECTION

In this section, we provide the full prompt with few-shot reasoning example for our proposed action selection in Sec 3.3.

## C.1 SYSTEM PROMPT

---

**System Prompt**

You are an expert in evaluating the performance of a web navigation agent. The agent is designed to help a human user navigate a website to complete a task. Given a user's task, a meta-plan to complete the task (i.e., the ideal step-by-step guidelines a successful agent should follow to complete the task), current webpage state, and multiple candidate actions that can be taken at current state, your job is to select the best action that would most effectively help complete the task.

Here's the information you'll have:

- **User's objective:** This is the task you're trying to complete.
- **Current URL:** This is the page you're currently navigating.
- **Current screenshot:** A screenshot of the current webpage, with each interactable element assigned a unique numerical id.
- **Current Observation:** Lists the IDs of all interactable elements in the format `[id] [tagType] [text content]`. `tagType` is the type of the element, such as button, link, or textbox.
- **Open tabs:** The tabs you have opened.
- **Previous actions:** History of actions that you have performed. It may be helpful to track your progress.
- **Previous screenshots:** History of previous screenshots leading up to the current state by previous actions history. It may be helpful to track your progress.
- **Meta-plan:** A structured high-level guideline outlining the logical sequence of steps you should consistently follow to effectively solve any given task.
- **Target actions:** The set of candidate target actions that can be performed at the current webpage state.

**The types of target actions:**

- *Page Operations:* `click [id]`, `type [id] [content]`, `hover [id]`, `scroll [down]`, etc.
- *Tab Management:* `new_tab`, `tab_focus [index]`, `close_tab`
- *Navigation:* `goto [url]`, `go_back`, `go_forward`
- *Completion:* `stop [answer]`

**Rules to Follow:**

1. Given multiple candidate actions that could be taken at the current state, your job is to compare all the action candidates and select the best one that would most effectively move the user toward completing the task, specifically by considering how far it would move the user forward along the meta-plan (i.e., the ideal trajectory).

2. Follow the examples to reason step by step and then select the best action.

3. You should provide reasoning that explains why the selected action is better than the other candidates, considering the user's task, the meta-plan, and the history of actions and screenshots.

---

## C.2 FEW-SHOT REASONING EXAMPLES

---

**Few-shot Reasoning Examples**

**## Inputs**

**Objective:** Buy the least expensive red blanket (in any size) from "Blankets & Throws" category.

---

**Current URL:** `http://_SHOPPING_DOMAIN_/home_kitchen.html`
**Current screenshot:**

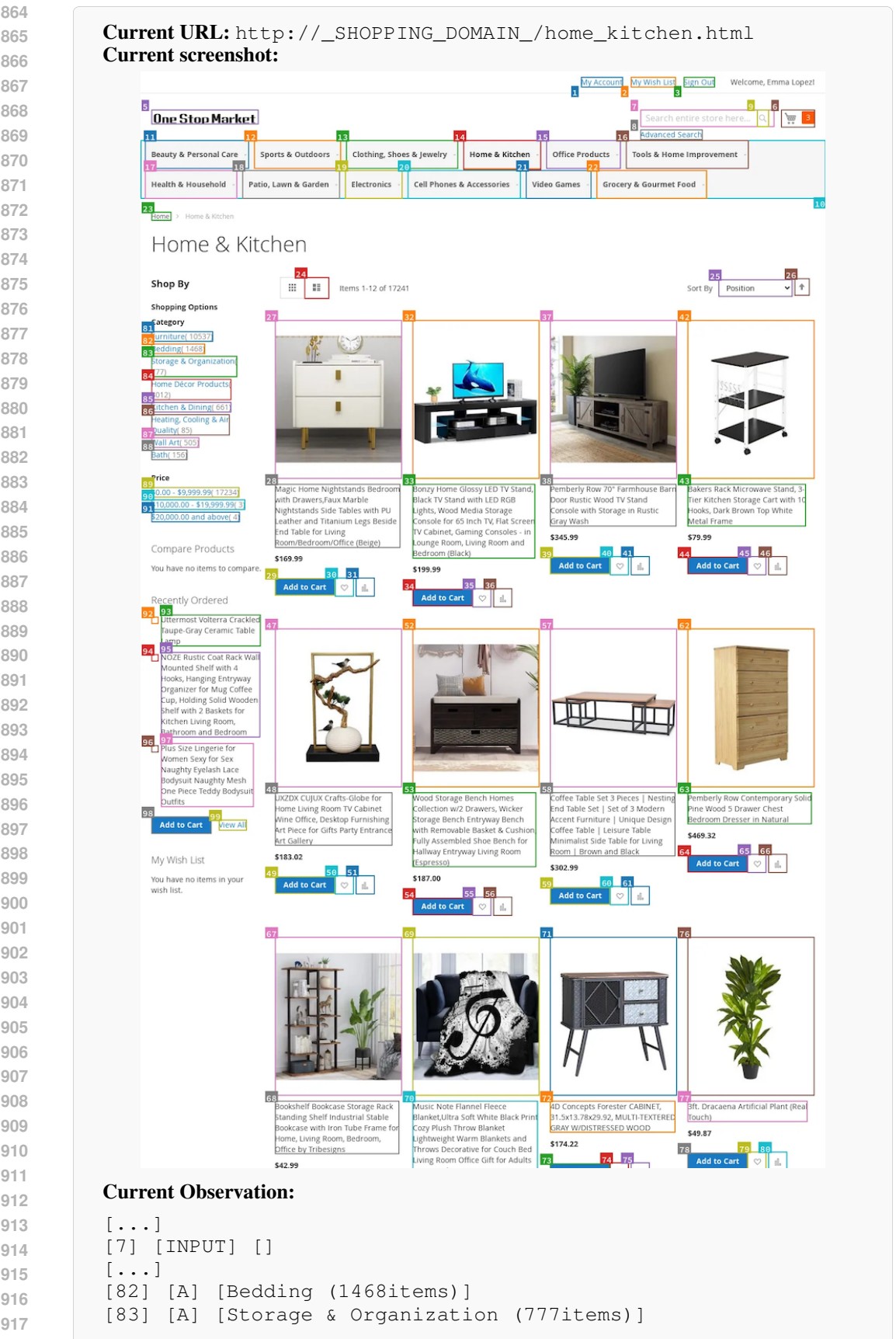

**Current Observation:**

```
[...]
[7] [INPUT] []
[...]
[82] [A] [Bedding (1468items)]
[83] [A] [Storage & Organization (777items)]
```

```
[...]
```

**Open tabs:** [Shopping]
**Previous actions:** None
**Previous screenshots:** None
**Meta-plan:**

- 1. Navigate to Shopping Website: Open the shopping website where the purchase will be made.

- 2. Access 'Blankets & Throws' Category: Navigate to the 'Blankets & Throws' category from the homepage.

- 3. Filter and sort: Apply a filter to show only red blankets (if available) and sort the results by price in ascending order.

- 4. Select Least Expensive Blanket: Choose the least expensive red blanket from the sorted list.

- 5. Add to Cart and Purchase: Add the selected blanket to the cart and proceed to complete the purchase by providing necessary payment and shipping information.

**Target actions:**

- 1. `click [82]` where [82] is [A] element with content [Bedding (1468items)]

- 2. `type [7] [blankets & throws\n]` where [7] is [INPUT] element with content []

---

## Output

**Reasoning:**

- *Step 1 (Task Analysis):* The user's task is to buy the least expensive red blanket (any size) from the 'Blankets & Throws' category.

- *Step 2 (Current Progress Analysis):*

    ✓ 1. Navigate to Shopping Website — The user is already on the homepage.
    ✗ 2. Access 'Blankets & Throws' Category — This is the next required step.

    The agent must first access the appropriate product category ('Blankets & Throws') before applying color filters or sorting by price.

- *Step 3 (Action Candidate Evaluation):*

    – Candidate 0: `click [82]` — Clicks on 'Bedding (1468 items)'
        ✓ Most logical next step, as 'Blankets & Throws' would likely be a subcategory within 'Bedding'.
        ✓ Aligns directly with the second step of the meta-plan.
    – Candidate 1: `type [7] [blankets & throws\n]`
        ✗ Search may result in noisy or incomplete listings, which can interfere with proper filtering and sorting.
        ✗ May not guarantee a categorized listing specific to 'Blankets & Throws'

**Selected Action:** Since the next upcoming step in the meta-plan is navigating via category links and that 'Blankets & Throws' is likely a subcategory of 'Bedding', Candidate 0 is the more reliable and aligned action. Thus, Candidate 0: `click [82]` is the correct action.

## C.3   CURRENT INPUTS

---
**Current Inputs**

**Objective:** [TASK INSTRUCTION]
**Current URL:** [CURRENT URL]
**Current screenshot:** [CURRENT WEBPAGE SCREENSHOT IMAGE]
**Current Observation:** [CURRENT OBSERVATION]
**Open tabs:** [OPEN TABS]
**Previous actions:** [PREVIOUS ACTIONS]
**Previous screenshots:** [PREVIOUS SCREENSHOTS]
**Meta-plan:** [META-PLAN]
**Target actions:** [TARGET ACTIONS]

---

## D   DYNAMIC REPLANNING OF META-PLAN

In this section, we provide the full prompt for dynamic replanning of the meta-plan. Instead of using the static meta-plan generated from Appendix A, the meta-plan is updated at every time step using below prompt, allowing the policy model to generate actions corresponding to the updated meta-plan.

---
**System Prompt**

You are an expert web navigation planner. You will be given the following inputs:

- **User's objective:** This is the task you're trying to complete.

- **Current screenshot:** A screenshot of the current webpage, with each interactable element assigned a unique numerical id.

- **Current Observation:** Lists the IDs of all interactable elements in the format `[id] [tagType] [text content]`. `tagType` is the type of the element, such as button, link, or textbox.

- **Previous actions:** History of actions that you have performed. It may be helpful to track your progress.

- **Previous screenshots:** History of previous screenshots leading up to the current state by previous actions history. It may be helpful to track your progress.

- **Meta-plan:** A structured high-level guideline outlining the logical sequence of steps you should consistently follow to effectively solve any given task.

**Actions you can perform:**

- *Page Operations:* `click [id]`, `type [id] [content]`, `hover [id]`, `scroll [down]`, etc.

- *Tab Management:* `new_tab`, `tab_focus [index]`, `close_tab`

- *Navigation:* `goto [url]`, `go_back`, `go_forward`

- *Completion:* `stop [answer]`

**Your job is to do the following:**

1. **Identify the Current Progress Status:** Based on the previous history and the current web page screenshot, identify the last fully completed stage in the meta-plan to assess how far the agent has progressed.

2. **Self-Verify the Next Stage**
   Evaluate whether the next incomplete stage in the meta-plan satisfies the following conditions based on the current observations:

   - ✓ **Executable**: Can this stage be completed by interacting with the elements currently listed in the observation?

---

- ✓ **Essential**: Is this stage necessary for achieving the user's objective?

If the next stage does not satisfy these conditions:

- **Revise** the plan by removing or updating the invalid stage.
- **Adjust** the subsequent stages as needed to ensure the meta-plan still leads to successful completion of the user's objective.

Otherwise, if the next stage is still valid and executable, keep it and all following stages unchanged.

**Output Format:**
Format your response as a JSON array of stage objects without additional commentary.
Example:

```
[
    {
        "stage_name": "Search",
        "description": "Search for the target product
        using appropriate keywords"
    },
    {
        "stage_name": "Price Comparison",
        "description": "Compare prices, potentially
        using sorting or filtering functionality"
    },
    ...
]
```

**Current Inputs**

**Objective:** [TASK INSTRUCTION]
**Current screenshot:** [CURRENT WEBPAGE SCREENSHOT IMAGE]
**Current Observation:** [CURRENT OBSERVATION]
**Previous actions:** [PREVIOUS ACTIONS]
**Previous screenshots:** [PREVIOUS SCREENSHOTS]
**Meta-plan:** [META-PLAN]

# E  QUALITATIVE RESULTS

## E.1  ACTION GENERATION

In Figure 6, we visualize the agent's trajectory of greedy reasoning and progress-aware reasoning during the action generation process in Sec. 3.2, respectively. Notably, although the agent has already found the ground truth target item, greedy reasoning (Gu et al., 2024) oscillates without completing the task. In contrast, our progress-aware reasoning successfully terminates by noticing that the last stage of the meta-plan has already been completed.

## E.2  ACTION SELECTION

In this section, we visualize qualitative examples of the action selection process by the conventional action value function (Gu et al., 2024) and our progress-aware action evaluator. In Figure 7, the conventional value function greedily assigns a higher value to the stopping action, prematurely terminating the task despite the objective (i.e., navigating to the detailed view of the identified item) not being fulfilled. In contrast, our proposed action evaluation selects the appropriate `click` action rather than the premature stop action, based on explicitly focusing on the next remaining step of the meta-plan (i.e., clicking the identified item), leading to the successful task completion.

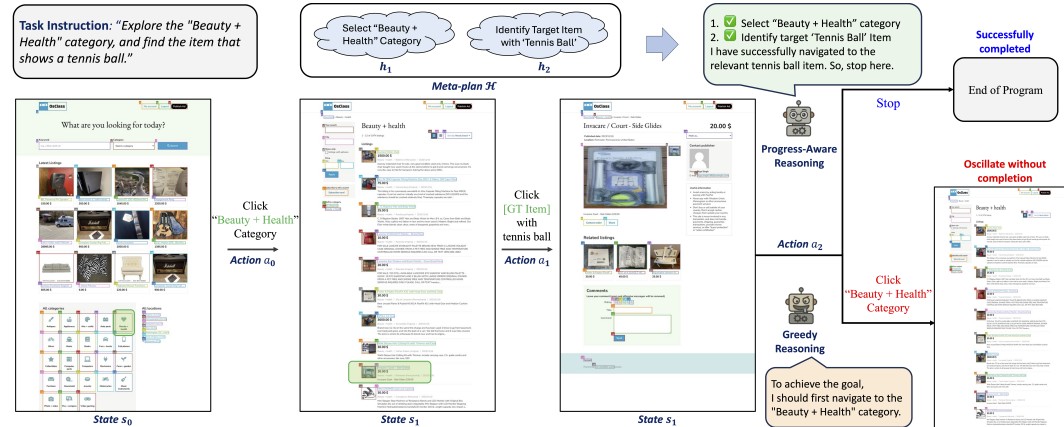

Figure 6: The trajectory of greedy reasoning (bottom; Yao et al. (2023); Gu et al. (2024); Koh et al. (2024b)) and progress-aware reasoning (top), where the agent should finalize the item search task.

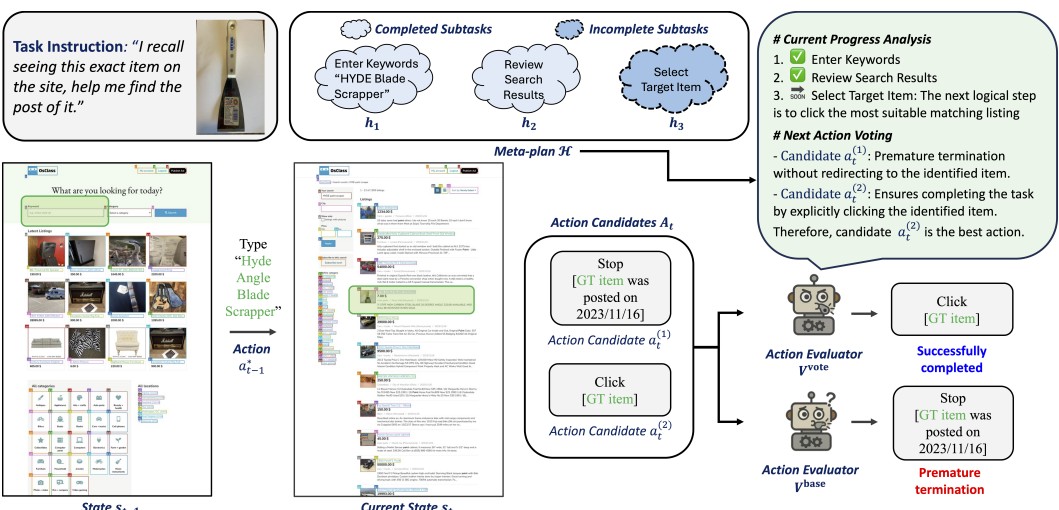

Figure 7: Action selection by conventional value function (bottom; Gu et al. (2024); Koh et al. (2024b)) and our progress-aware action evaluator (top), where the agent should finalize the item search task.

Also, in Figure 8, we note that our proposed evaluator also effectively resolves the invalid action repetition issue, where the conventional action evaluator repeatedly selects the scroll down action although the page is already reached to the bottom. However, unlike the conventional evaluator, our proposed action evaluator selects more effective action (i.e., turning to the next page) that enables further search, deviating from the invalid action loop by focusing on the next remaining subgoal in the meta-plan.

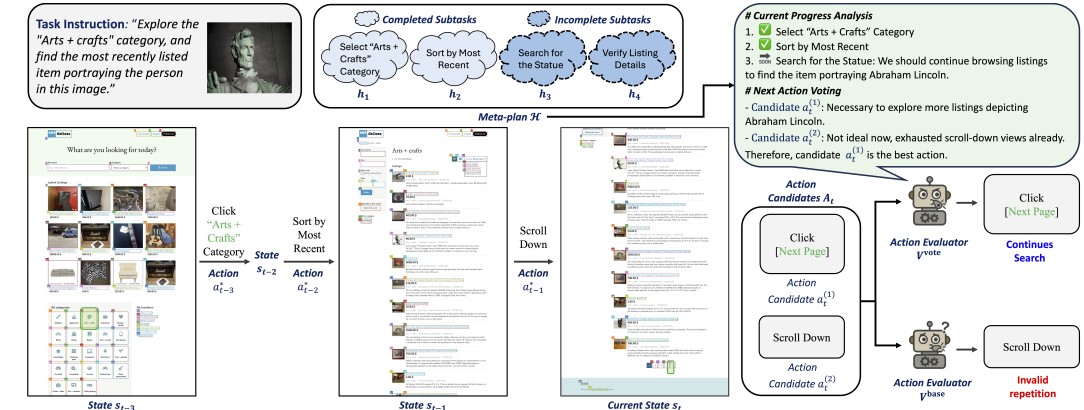

Figure 8: Action selection by conventional value function (bottom; Gu et al. (2024); Koh et al. (2024b)) and our progress-aware action evaluator (top), where the agent should continue searching for the target item on the next pages.

# F   RESOURCE EFFICIENCY

In Table 5, we further analyze the efficiency of MAPLE in terms of computational resource overhead. The results reveal that the previous SOTA frameworks, such as WebDreamer and TreeSearch, induce either heavy API cost or long execution time due to their additional computational components, including simulation-based rollout (Gu et al., 2024) and backtracking (Koh et al., 2024b) mechanisms. Without relying on these computationally heavy routines, our proposed method largely reduces resource overhead while achieving even higher task success rates. Specifically, MAPLE requires less than half the API cost and latency compared to WebDreamer and TreeSearch, respectively, while representing a large headroom for the success rate.

Table 5: Resource efficiency comparison with SOTA frameworks, WebDreamer and TreeSearch. We report the average of task completion time, API cost (calculated by the number of input and output tokens spent), and task success rate for each framework. For fair comparison, we used GPT-4o for all the frameworks and set the maximum number of time steps to 30.

| Method | Time (sec) ↓ | API Cost ($) ↓ | Classifieds SR (%) ↑ |
|---|---|---|---|
| WebDreamer (Gu et al., 2024) | 665 | 2.61 | 23.2 |
| TreeSearch (Koh et al., 2024b) | 1043 | 1.29 | 29.1 |
| **MAPLE (Eq 6, 8)** | **500** | **1.12** | **32.1** |

