# OpenReview forum: "Web Agents Are Still Greedy: Progress-Aware Action Generation and Selection via Meta-Plan"
_ICLR.cc/2026/Conference — ICLR 2026 Conference Withdrawn Submission_

### Official Review · Reviewer_ncL2 · 2025-10-28

**Soundness:** 3
**Presentation:** 3
**Contribution:** 3
**Rating:** 6
**Confidence:** 4

**Summary:**

This paper addresses the problem of “greedy reasoning” in web agents, where models make short-sighted decisions without tracking task progress. The authors propose MAPLE, a framework that introduces meta-plan–guided reasoning to make web agents more progress-aware. MAPLE generates a high-level meta-plan to outline task structure, uses LLM-based reasoning to assess candidate actions, and employs a voting mechanism to select the most consistent next step. Experiments on the VisualWebArena benchmark show that MAPLE outperforms existing web agent baselines, and ablation studies confirm the contribution of its meta-planning and action selection modules.

**Strengths:**

1. The paper introduces a meta-plan–guided reasoning framework for action generation and selection, enabling the model to learn from previous interactions and track task progress to inform decision-making. Experimental results demonstrate that the approach can function in a plug-and-play manner, making it compatible with existing web agent frameworks and enhancing their overall performance.

2. The proposed method achieves consistent performance improvements over several baselines on the VisualWebArena dataset. In addition, ablation studies validate the effectiveness of key design components and highlight the benefits of the proposed meta-planning mechanism.

3. The paper is well-written, clearly motivated, and easy to follow, presenting its contributions and experimental results in a structured and coherent way.

**Weaknesses:**

1. The model’s performance appears highly dependent on the quality of the LLM-generated meta-plan. It remains unclear how the framework would perform if a weaker multimodal LLM were used—particularly one that produces incomplete or poorly structured plans (see my first question). Some analysis or discussion of this dependency would strengthen the paper.

2. The experiments are conducted exclusively on the VisualWebArena dataset. Given that the proposed method is relatively general, it would be valuable to explore more dynamic multimodal environments to assess how well the approach generalizes to scenarios with different visual or interaction dynamics.

3. The framework involves multiple LLM calls, including those for meta-plan generation, candidate action reasoning, and voting, which may introduce nontrivial computational overhead. A discussion or quantification of the trade-off between performance gains and inference cost would help clarify the method’s practical feasibility.

**Questions:**

1. The proposed MAPLE framework relies heavily on LLM-generated meta-plans to guide reasoning and action selection. It would be great if the authors could clarify how sensitive the model’s performance is to the quality of these meta-plans? In particular, how would MAPLE perform if the underlying multimodal LLM produces weaker or poorly structured plans—for example, when using a smaller or less capable model? Have the authors conducted any experiments or ablations to assess this dependency?

2. The experiments are conducted exclusively on the VisualWebArena dataset, but the proposed approach appears general enough to apply to other multimodal planning or reasoning scenarios. Have the authors considered evaluating MAPLE in more dynamic or diverse multimodal environments (e.g., tasks with different visual or interaction dynamics)? If not, could the authors provide some discussion or anticipation of how they expect the model to generalize to such settings?

3. According to Tables 3 and 4, the model shows only modest improvements on the more challenging reasoning tasks. How do the authors interpret this performance drop? Is the decrease primarily due to failures in meta-planning (e.g., incomplete or inaccurate plans) or in the action selection process?

4. According to my last weakness, have the authors analyzed the additional computational overhead associated with these components? It would be helpful to discuss the trade-off between performance gains and inference cost, especially for real-world or large-scale deployment.

---

### Official Review · Reviewer_oYyg · 2025-10-30

**Soundness:** 3
**Presentation:** 3
**Contribution:** 2
**Rating:** 4
**Confidence:** 3

**Summary:**

This paper proposes a framework designed to enhance the reasoning reliability of LLM-based web agents by making them progress-aware. The authors argue that current agents suffer from greedy reasoning, focusing only on immediate goals without tracking what subtasks have been completed or remain. To address this, the proposed framework called MAPLE integrates a meta-plan which decomposes a task instruction into several high-level subtasks, and a progress vector that tracks subgoal completion over time. At each step, the agent generates action candidates conditioned on the meta-plan and selects the goal-directed action through a meta-plan–guided voting mechanism. Experiments show substantial improvements on long-horizon, reasoning-intensive tasks. Ablation and efficiency analyses confirm that MAPLE achieves both higher task success and lower resource overhead compared to prior frameworks.

**Strengths:**

* Important topic of research: The paper tackles a prominent challenge of failure to maintain coherence in long-horizon reasoning for AI agents. By attributing common error patterns such as premature termination, step skipping, and action repetition to greedy step-wise decision-making, the authors provide a clear rationale for introducing an explicit progress-tracking mechanism.
* Plug and play technology: The proposed methodology is simple yet broadly applicable. The two-stage framework of meta-plan generation and step-by-step action selection requires no fine-tuning or additional training. Both stages rely entirely on prompting with in-context examples, making the method lightweight and easily integrable with existing agents. The authors empirically confirm this plug-and-play adaptability and report consistent performance gains when the proposed framework is layered on top of base systems.
* Methodological novelty: The proposed framework introduces a structured meta-plan that decomposes a user’s instruction into high-level subtasks. This explicit representation allows the agent to reason about what has been done and what remains. Figure 1 and Appendix A clearly illustrate how this meta-plan grounding guides more coherent trajectories compared to short-sighted reasoning.
* Rigorous experimentation: The authors evaluate the framework on the VisualWebArena benchmark, covering three distinct website environments with 910 tasks. The experiments include solid comparisons against previous arts like ReAct, ICAL, AdaptAgent, GenericAgent, WebDreamer, and TreeSearch. The proposed approach achieves the best success rate across all domains and demonstrates both generalization and robustness.
* Qualitative analysis: Through visualizations across Fig. 6 - 8, the authors present interpretable examples of the proposed framework's trajectories versus baseline agents. These examples demonstrate how progress tracking and meta-plan alignment eliminate redundant actions and recover from irrelevant behaviors.
* Meta-cognitive framework: The proposed framework's emphasis on self-referential task monitoring bridges reasoning and planning which is a step toward more autonomous, meta-cognitive agents. Its progress-tracking approach can potentially extend beyond web agents to embodied or interactive systems, making it a meaningful conceptual contribution to the broader LLM-agent literature.

**Weaknesses:**

* Unsubstantiated claim that prior agents are greedy: The paper’s central motivation that previous agents fail due to greedy reasoning is asserted but not rigorously supported by backup references nor empirically demonstrated. The authors cite qualitative examples (e.g., Figure 1) but provide no behavioral metrics or quantitative evidence that even structured methodologies like TreeSearch and WebDreamer act greedily. How is greedy reasoning operationally defined or measured? Could other factors like environment drift, action grounding errors etc. explain the same failures?
* Reliability of the progress-completion function: The completion judgment is crucial for implementing progress-awareness yet its operationalized mechanism is evaluated only qualitatively. Without quantitative accuracy or consistency analysis, it remains unclear how reliable this binary reasoning is. Reporting completion accuracy or self-consistency scores would make the progress-aware claim more scientifically grounded.
* Fixed-length meta-plan: The choice of five subtasks per instruction appears empirically convenient but theoretically arbitrary. Tasks of varying complexity likely require different granularity, and the paper provides no rationale or discussion on this issue. Would the performance degrade when tasks require more than five conceptual steps?

**Questions:**

Please refer to the weaknesses part.

---

### Official Review · Reviewer_D4kd · 2025-10-31

**Soundness:** 3
**Presentation:** 3
**Contribution:** 2
**Rating:** 4
**Confidence:** 5

**Summary:**

This paper proposes MAPLE, a prompting method for action generation and selection. At inference time, a meta-policy model decomposes the objective into a sequence of subtasks and uses this to guide the agent during training. This meta-plan is used as a guide for the agent to select the plans. Furthermore, the policy model will generate multiple action candidates at each time step and majority voting is used to select the final action. The paper compares their method against multiple prior work and achieves strong performance on VisualWebArena benchmark.

**Strengths:**

The method is clear and simple to understand and Section 3.2 does a good job at explaining the main idea. The implementation details are thorough and the results shown in Table 1 are clear and show that MAPLE outperforms the stated baselines. Further ablations on adding the method to other baselines, as well as ablations on PAR and dynamic replanning, and their hyperparameters for inference time scaling are insightful and clear. Appendix contains plenty of different examples and prompts to promote reproducibility.

**Weaknesses:**

The main weakness of the paper is that it is solely focused on 1 simulated web agent benchmark. Paper would be significantly strengthened if the paper could target other web datasets such as WebArena [1], WebArena-lite [2] , WebVoyager [3] , Online-Mind2Web [4] , etc.
Other benchmarks that could be used are GUI based benchmarks such as OSWorld [5]  or AndroidWorld [6] . One does not have to do every single one of these benchmarks, but having at least 1 or 2 other datasets where this technique is evaluated would be more convincing.

Furthermore, this idea of using a meta-plan to keep track of progress and subgoals is not a new idea, not even with web agents [8, 9]. The sole added novelty seems to be in using test-time scaling to sample multiple actions per step and checking them against the meta plan.

Paper is also missing comparison with ExACT [7], which also uses GPT-4o as a base model. It achieves a 33.7% performance on VisualWebArena, which is on-par with the best result in Table 1.

Section 4.2 and Table 4 seems to claim that dynamic replanning of the meta-plan seems to lessen the performance of the model, which is an interesting ablation if true. It would be very insightful to run this agent again with no Action Gen. w/ Meta Plan, but with Dynamic Replanning turned on to see how that model performs. This result seems to indicate that this technique is detrimental when using test-time scaling of the Action Gen. w/ Meta Plan.

As a further note, while the results in Table 3 and 4 are interesting, this analysis is restricted to just the Classifieds subset of VWA. It would be a more robust and stronger result if it was applied to the the entirety of VWA.


[1] Zhou, Shuyan, et al. "WebArena: A Realistic Web Environment for Building Autonomous Agents." NeurIPS 2023 Foundation Models for Decision Making Workshop.

[2] Liu, Xiao, et al. "VisualAgentBench: Towards Large Multimodal Models as Visual Foundation Agents." The Thirteenth International Conference on Learning Representations.

[3] He, Hongliang, et al. "WebVoyager: Building an End-to-End Web Agent with Large Multimodal Models." Proceedings of the 62nd Annual Meeting of the Association for Computational Linguistics (Volume 1: Long Papers). 2024.

[4] Xue, Tianci, et al. "An illusion of progress? assessing the current state of web agents." arXiv preprint arXiv:2504.01382 (2025).

[5] Xie, Tianbao, et al. "Osworld: Benchmarking multimodal agents for open-ended tasks in real computer environments." Advances in Neural Information Processing Systems 37 (2024): 52040-52094.

[6] Rawles, Christopher, et al. "AndroidWorld: A Dynamic Benchmarking Environment for Autonomous Agents." The Thirteenth International Conference on Learning Representations.

[7] Yu, Xiao, et al. "ExACT: Teaching AI Agents to Explore with Reflective-MCTS and Exploratory Learning." The Thirteenth International Conference on Learning Representations.

[8] Kim, Minsoo, et al. "Rada: Retrieval-augmented web agent planning with llms." Findings of the Association for Computational Linguistics ACL 2024. 2024.

[9] Wang, Haoyu, et al. "Devil’s Advocate: Anticipatory Reflection for LLM Agents." Findings of the Association for Computational Linguistics: EMNLP 2024. 2024.

**Questions:**

Could the authors provide a result from 1-2 of the benchmarks listed in the weaknesses section above? If sticking to web tasks, having WebArena-lite as an additional benchmark (which has some overlap with VWA) and/or WebVoyager, which evaluates the model on real world websites and environments would provide much stronger evidance.

Furthermore, I would love to see ExACT added to the table and whether adding Maple to ExACT would also improve performance, as in Table 2.

It would also be great if the authors could provide some of the other experiments listed in the weaknesses section above related to Section 4.2.

---

### Official Review · Reviewer_akfi · 2025-10-31

**Soundness:** 2
**Presentation:** 3
**Contribution:** 1
**Rating:** 2
**Confidence:** 4

**Summary:**

The paper identifies that a key limitation of web-agents is that they take a greedy action at every step of a trajectory which might not be the most optimal action for the overall success of the goal. To remedy this, they propose an add-on method called MAPLE (MetA PLan guided action generation and sElection) which allows a web-agent to create a high-level, sequential plan for fulfilling a task. A meta-plan allows the agent to track its progress during its trajectory and not fall into sub-optimal behaviors. Via this method, the authors demonstrate higher success rates (SR) of 33.3% on Visual WebArena benchmark compared to other contemporary methods like WebDreamer (23.2%) and TreeSearch (26.2%).

**Strengths:**

1. The proposed MAPLE framework has the advantage of being a lightweight method which can be added on to existing agent frameworks to increase their reliability. Experiments in the paper show that it is helpful to perform action candidate generation and selection based on meta-planning to web agent designs.
2. The proposed approach achieves gains over existing agent frameworks WebDreamer and TreeSearch, which are used as baselines in this paper, on VisualWebArena (Classifieds).
3. Ablative studies and analysis help the reader understand the contributions of different techniques in the proposed framework, which is reasonably comprehensive.

**Weaknesses:**

1. Existing agent frameworks [1, 2, 3, 4, 5, 6] have explored the technique of hierarchical and meta-planning, as well as multiple action generation and best-action selection extensively. As claimed by the paper as well (L128), this work differentiates itself by adding progress awareness to the agent by tracking a meta-plan, yet it fails to discuss the connection of this approach to prior work such as [7, 8, 9]. In addition, in fact approaches like WebPilot also track finished sub-tasks which further reduces the novelty of MAPLE.
2. The central claim of “Due to the lack of such progress awareness, they often suffer from suboptimal behaviors such as skipping essential key steps for solving the long-horizon tasks” in L126 is not sufficiently backed by empirical evidence.
3. More benchmarks apart from Visual-WebArena, like WebVoyager/WebArena can be used in the paper to demonstrate whether the MAPLE is can be generalized to diverse web navigation tasks.
4. The paper title claims that the motivation of the work is that web-agents act greedily during a trajectory.  However, the proposed method of meta-planning is greedy nonetheless, even though it is greedy at a planning level instead of the per-action level. MAPLE produces a best plan in one-shot by assuming how general web-pages work. However the main complexity of real-life websites comes from their dynamic nature and the need for adaptability and backtracking while traversing websites (For eg, a WebArena task like “What is the top-1 best-selling product in 2022” the meta-plan might start with `navigate to the page containing product listings` but depending on the website mechanism, they might have an alternative correct first step like `go to the bestsellers page`. This indicates that the meta-plan needs to be revised, the planned step might not work when actual grounding is attempted, the website state itself might change due to certain write actions taken during the meta-plan etc. The paper does not show empirical evidence for how these are addressed by MAPLE.
5. Questions of whether this approach will adapt to long-horizon, and non-linear goals (eg, tasks/workflows that require loops or conditionals) can also be addressed in the paper to establish the practical usability of this technique.

References included in paper:
[1] TPTU: Large Language Model-based AI Agents for Task Planning and Tool Usage (https://arxiv.org/abs/2308.03427)

[2] Plan-and-Act: Improving Planning of Agents for Long-Horizon Tasks
 (https://arxiv.org/abs/2503.09572)

[3] Tree Search for Language Model Agents (https://arxiv.org/abs/2407.01476)

References not included in paper:
[4] Language Agent Tree Search Unifies Reasoning Acting and Planning in Language Models (https://arxiv.org/abs/2310.04406)

[5] WebPilot: A Versatile and Autonomous Multi-Agent System for Web Task
Execution with Strategic Exploration (https://arxiv.org/pdf/2408.15978)

[6] AdaPlanner: Adaptive Planning from Feedback with
Language Models (https://arxiv.org/pdf/2305.16653)

[7] Globally Coherent Text Generation with Neural Checklist Models. (https://aclanthology.org/D16-1032.pdf)

[8] PC-Agent: A Hierarchical Multi-Agent Collaboration Framework for Complex Task Automation on PC (https://arxiv.org/abs/2502.14282)

[9] Self-Monitoring Navigation Agent via Auxiliary Progress Estimation (https://arxiv.org/abs/1901.03035)

**Questions:**

Suggestions:
1. The paper could differentiate their novelty and innovation compared to other works discussed in the Related Works. More related works using planning frameworks could also be mentioned to clarify that hierarchical planning and multi-action generation and selection is a widely used technique in web agent literature.
2. Adding baseline results with more frontier foundational and computer-use models from Anthropic and newer models from OpenAI (GPT4.1/GPT-5) would be helpful. This will help determine whether a multi-step planning framework like MAPLE is required by computer-use models which perform the reasoning and backtracking necessary for web tasks step-by-step.
3. The paper could have tested the generalizability of MAPLE to other small language model policy models to inform readers who want to adopt the technique to open-source  language models.
Questions:
1. Why are the number of subtasks generated in meta-plan (K) limited to 5? What is the recommendation for setting this value for more complex tasks?
2. How well does the action candidate generation pipeline identify which subtasks of the meta-plan are completed and which are not?
3. GPT-4o is used as the action value function for determining best action selection. Did the authors experiment with any other value functions? Do we have quantitative results to show how accurate the value function is at best action selection?
4. In L321, how are the in context examples selected for the voting model $V^{vote}$?
5. Can we clarify what is the baseline method in L428?
6. What is the variance in the experimental results? Were multiple runs used to compute the success rates on VisualWebArena?

---

### Official Review · Reviewer_XfJ2 · 2025-10-31

**Soundness:** 3
**Presentation:** 2
**Contribution:** 2
**Rating:** 2
**Confidence:** 3

**Summary:**

The paper introduces MAPLE, a training-free plug-in that gives web agents explicit progress awareness. It first drafts a short meta-plan of high-level steps and, at each step, infers a binary progress state to focus behavior on the next unmet subgoal.
The agent proposes multiple action candidates and a voting module selects the one that best advances that subgoal.
MAPLE yields consistent gains on a standard web-browsing benchmark without additional training, and it reduces runtime and API cost while integrating easily into existing prompt-based agents.

**Strengths:**

- The paper points to a clear cause of failure in web agents (missing progress awareness) and puts a meta-plan, a simple binary progress state, and a voting step together as a small plug-in that is new in this web setting and removes a common limit of prompt-only agents.

- The method is consistent end to end, uses the same backbone as planner, actor, and judge to keep controls fair, and the experiments compare against recent systems with key ablations that match the claims.

- The pipeline is easy to follow (make a short plan, check progress, propose actions, vote), and the paper shows concrete examples and figures that make each step clear.

- It needs no training and drops into existing agents with little work, improving success and cost, which makes it practical for real deployments and a simple baseline for future hierarchical agents.

**Weaknesses:**

* **Scope and transfer to trained agents** — The paper only tests a zero-training, prompt-based setup where the same model plans, acts, and judges. In practice, many web agents are trained for a given environment with imitation learning or RL and already learn hierarchy and progress. It is unclear if MAPLE still gives clear gains there, or if the external meta-plan and voting might overlap with or even constrain what the trained policy has learned. This will determine the level of contribution of the idea. In addition, with smaller or weaker backbones the policy may rarely propose on-track actions, so voting has little to choose from. Similarly, in more complex environments with longer flows and tighter constraints, the effect is also unknown. This gap needs evidence and discussion.

* **Per-step re-planning underperforms** — The paper itself reports that re-planning at every step hurts on harder tasks. This may happen because simple workflows do not need a plan at all (the extra planning adds overhead), while very complex workflows can make the initial plan wrong and force frequent changes. Frequent re-planning can also reset context, break progress, cause oscillation between subgoals, and add token/time cost; together these effects may make the method struggle when plans must change often.

* **Path efficiency metrics missing?** — Results report success, time, and API cost, but not average steps, path length, or redundancy, so it is hard to tell if paths are actually shorter; there also does not seem to be an ablation that targets this question.

* **Sampling variance and statistical stability** — The approach relies on stochastic sampling and multi-round voting, which adds randomness, yet the paper does not show multi-seed means and variance or statistical significance tests, so stability across runs is unclear.

* **Seed sensitivity** — The pipeline depends on templates and few-shot examples; reproduction can drift and results may swing.

**Questions:**

* **Can MAPLE improve agents that are already trained with IL or RL?**
  I'm considered about whether the external meta-plan and voting still add clear gains or interfere with what the policy has learned.

* **What happens with smaller or weaker backbones?**
  If the base model rarely proposes on-track actions, will voting still help?

* **How does MAPLE behave in more complex environments with longer flows and tighter constraints?**
  Full experiments may be heavy, a short analysis or reasoned discussion is fine to explain expected behavior and limits in such settings.

* **Can you report path-efficiency metrics (average steps, path length, redundancy)?**
  Success and time are useful, but path metrics would show if the agent truly takes shorter and cleaner routes.

* **Can you provide multi-seed results and basic significance tests?**
  Since decoding and voting are stochastic, mean±std over seeds (and simple tests) would show stability and reduce concerns about variance.

* **Can progress be more than binary (for example a soft score or evidence count)?**

---

### Official Review · Reviewer_dBmT · 2025-11-01

**Soundness:** 4
**Presentation:** 3
**Contribution:** 2
**Rating:** 4
**Confidence:** 4

**Summary:**

This paper proposes MAPLE, a plug-and-play module that explicitly tracks what has been completed and what remains, addressing the  failure caused by greedy next-action reasoning. It first decomposes the user instruction into a high-level meta-plan, and uses meta-plan to generate candidate actions, then selects the next action by majority voting, giving LLM-based web agents progress awareness that reduces skipped steps, oscillations, and premature termination. Experiments on VisualWebArena show substantial gains over existing approaches, with success rate improvements.

**Strengths:**

The paper proposes decomposing user instructions into a meta-plan and leveraging historical states and actions to guide both action planning and selection, offering a creative and plug-and-play solution to the problem of greedy next-step reasoning in web agents. Empirically, MAPLE demonstrates strong performance: on the VisualWebArena benchmark, it consistently outperforms competitive baselines, achieving up to 89% relative improvement in success rate. Detailed ablations validate the effectiveness of each component, and robustness is shown across varying task difficulties. The paper is well-structured, the methodology clearly articulated, and the approach generalizes across agent architectures, enhancing reliability and efficiency in long-horizon web tasks while addressing the limitations of prior value-function heuristics.

**Weaknesses:**

The paper offers limited methodological innovation. First, using historical states and actions as decision context has been extensively studied in planning-related work (refer to literature [1]-[4] as examples), so the contribution here is incremental even if its effectiveness for web agents is empirically demonstrated. Second, the claimed contribution of “action selection with meta-plan” is largely implemented by prompting an LLM; although Equations (6)–(8) formalize the procedure, they do not constitute an independent methodological advance. Overall, the work delivers a practical improvement for a specific application, but its technical novelty and depth of insight fall short of ICLR’s acceptance standards.

[1] Chen, Shizhe, et al. "History aware multimodal transformer for vision-and-language navigation." Advances in neural information processing systems 34 (2021): 5834-5847.
[2] Jeon, Sujin, Suyeon Shin, and Byoung-Tak Zhang. "HAPFI: History-Aware Planning based on Fused Information." 2024 IEEE International Conference on Robotics and Automation (ICRA). IEEE, 2024.
[3] Chen, Siwei, Anxing Xiao, and David Hsu. "Llm-state: Open world state representation for long-horizon task planning with large language model." arXiv preprint arXiv:2311.17406 (2023).
[4] Zhai, Mingliang, et al. "Memory-Centric Embodied Question Answer." arXiv preprint arXiv:2505.13948 (2025).

**Questions:**

Q1. Task Definition and Clarity.
It remains unclear whether the actions executed by the agent are selected from a fixed, pre-defined action set or generated as open-ended language descriptions. A brief introduction to this task or a dedicated paragraph clarifying this aspect would help make the paper more self-contained and ensure that readers fully understand the goal and scope of the task.

Q2. Efficiency and Historical Information.
As shown in Figure 1, the proposed method effectively addresses failure cases caused by repeated actions. By leveraging historical information, the method should also have the potential to improve task efficiency, for instance by reducing the number of steps required to accomplish a user instruction. Experimental comparisons or analyses illustrating this aspect would further strengthen the paper’s empirical insights.

Q3. Decomposition of Instructions.
In Section 3.2, the input instruction is decomposed into at most five subtasks. While this upper bound is reasonable for simple instructions, it may be restrictive for more complex ones. In such cases, each resulting subtask could still resemble a new instruction for the web agent to accomplish. It would be helpful if the authors could discuss the potential solutions to this issue or/and whether there exists a mechanism to quantify the complexity level of the input instruction or the resulting subtasks.

Q4. Mathematical Formulation and Clarity.
In Section 3.2, the relationship between c_t (obtained from Eq. 5) and the predicted next action a_t (from Eq. 4) is unclear. Similarly, the definitions and roles of C(\cdot) and h_k^* are somewhat confusing. The authors are encouraged to revise this section by reorganizing the paragraph for clarity and presenting a more rigorous mathematical formulation.

Q5. Dependence on LLM/VLM Configurations.
Since the proposed method heavily relies on LLM/VLM components and hyperparameter settings (e.g., temperature, number of voting rounds), it would be valuable to test with a wider variety of LLM/VLM models to assess how model choice and parameter variations affect performance.

---

### Note · Authors · 2025-11-14

**Comment:**

Dear Program Committee and Reviewers,

After careful consideration, we have decided to withdraw our paper from the conference review process. We would like to express our gratitude for the constructive feedback, which has provided valuable insights into how we can enhance our research. We look forward to carefully implementing these suggestions and potentially resubmitting an improved version to a future venue.

Thank you once again for your time and consideration.

**Withdrawal Confirmation:**

I have read and agree with the venue's withdrawal policy on behalf of myself and my co-authors.